# Effect of Different Heat Treatments on the Quality and Flavor Compounds of Black Tibetan Sheep Meat by HS-GC-IMS Coupled with Multivariate Analysis

**DOI:** 10.3390/molecules28010165

**Published:** 2022-12-25

**Authors:** Jiqian Liu, Lijuan Han, Wenzheng Han, Linsheng Gui, Zhenzhen Yuan, Shengzhen Hou, Zhiyou Wang, Baochun Yang, Sayed Haidar Abbas Raza, Abdulaziz Faisal Saleh Alowais, Alaa Ahmed Alraddadi, Anwar Mohammed Alanazi

**Affiliations:** 1College of Agriculture and Animal Husbandry, Qinghai University, Xining 810016, China; 2College of Animal Science and Technology, Northwest A&F University, Yangling, Xianyang 712100, China; 3Safety of Livestock and Poultry Products, College of Food Science, South China Agricultural University, Guangzhou 510642, China; 4College of Medicine, Shaqra University, P.O. Box 13343, Riyadh 7396, Saudi Arabia; 5Laboratory Department, Alyamama Hospital, Ministry of Health, Riyadh 11451, Saudi Arabia

**Keywords:** black Tibetan sheep, heat treatment, meat flavor, pan-fried meat quality

## Abstract

There are limited reports about the effect of different heat treatments on the quality and flavor of Black Tibetan sheep meat. The current study examined the effect of pan-frying, deep-frying, baking, and boiling treatment on the quality of Black Tibetan sheep meat; the amino acid, fatty acid, and volatile flavor compounds (VFCs) were investigated by a texture analyzer, ultra-high-performance liquid chromatography (UHPLC), gas chromatography (GC), and headspace-gas chromatography-ion mobility (HS-GC-IMS). The key VFCs were identified through orthogonal partial least squares discrimination analysis (OPLS-DA), and variable importance projection (VIP) values. In addition, Pearson’s correlations between meat quality parameters and key VFCs were examined. The sensory scores, including texture, color, and appearance, of baked and pan-fried meat were higher than those of deep-fried and boiled meat. The protein (40.47%) and amino acid (62.93 µmol/g) contents were the highest in pan-fried meat (*p* < 0.05). Additionally, it contained the highest amounts of monounsaturated and polyunsaturated fatty acids, such as oleic, linoleic, and α-linolenic acids (*p* < 0.05). Meanwhile, pan-fried and deep-fried meat had higher amounts of VFCs than baked meat. The OPLS-DA similarity and fingerprinting analyses revealed significant differences between the three heat treatment methods. Aldehydes were the key aroma compounds in pan-fried meat. Importantly, 3-methylbutyraldehyde and 2-heptanone contents were positively correlated with eicosenoic, oleic, isooleic, linoleic, α-Linolenic, and eicosadiene acids (*p* < 0.05). To sum up, pan-fried Black Tibetan sheep meat had the best edible, nutritional, and flavor quality.

## 1. Introduction

Black Tibetan sheep (BTS), also known as black fur sheep in Guide, are a specialty of Guinan County, Hainan Tibetan Autonomous Prefecture, Qinghai Province, and a national geographical indication product of China. BTS lives in an environment of high altitude, low oxygen, and low temperature [1]. BTS meat is quite popular for its unique flavor, texture, and nutritional value. Previously, we reported that BTS meat has high nutritional value with high protein and amino acids content and low-fat content [2]. Concerning the sensory value, BTS meat is more tender, less fibrous, and chewy. Moreover, the meat quality (tenderness, flesh color, water retention, etc.) of one breed BTS is better than others such as the White Tibetan sheep. The current research mainly focuses on the effect of feed on the meat quality, rumen parameters, and growth performance of BTS [2,3], while the effect of different heat processing methods on the meat characteristics of BTS is largely unclear.

Heat processing of meat improves its flavor, texture, digestion characteristics, and shelf life [4]. For instance, meat stewing in brine enhances the quality and taste, hot-curing reduces fat while tightening the skin, and cooking makes the meat tender, thus increasing consumers’ acceptance and commercial value of meat products [5,6]. Secondly, the cooking process brings chemical changes, such as protein denaturation, lipid oxidation, degradation, Maillard reactions, etc., in meat products, which impact their nutritional value [7]. Therefore, it is important to select the appropriate meat cooking method. At present, the common mutton processing methods are steaming, boiling, stewing, deep-frying, baking, grilling, pan-frying, etc. [8]. Different cooking methods require different processing temperatures and times and therefore have different impacts on mutton. Wei Jia et al. [9] found that boiled and steamed goat meat has better tenderness and digestibility than roasted goat meat. It is believed that fat, which contributes to meat flavor and texture, is an important component of mutton and affects consumer acceptance [10]. Grinding, mixing, precooking, and storage of cooked meat increases fat oxidation causing the loss of meat’s nutritional and sensory value [11]. Additionally, higher polyunsaturated fatty acid (PUFAs) content increases lipid oxidation in meat [12]. Gravador et al. [13] found that grilling and sous-vide cooking increased the fatty acids content, especially n-3 fatty acids, in lamb meat.

Meat flavor is one of the most important sensory factors, which directly affects the consumer’s purchase decision. Cooked meat flavor is the result of non-volatile substances in fresh meat that undergo chemical reactions during thermal processing [14]. Roldán et al. [15] found that different temperatures and time combinations affected the flavor of vacuum-cooked lamb; a long cooking time and medium or high temperature produced the best meat flavor. The volatile flavor substances in cooked meat are mainly the products of the Maillard reaction and lipid degradation [15]. The fatty acid composition, intramuscular fat content, and oxidative stability determine the texture, juiciness, taste, and overall flavor of the meat. Aldehydes, alcohols, and ketones are the most volatile flavor substances in mutton [2], which mainly originate from fatty acids and their degradation products. Therefore, the flavor of cooked meat is largely related to the type and content of fatty acids. To our knowledge, no information has been published regarding the effect of different heat processing methods on the meat flavor of BTS. A study of the essential flavor-active compounds that originate from fatty acids and contribute to the perception of BTS meat flavor would help the development of this product. Accordingly, this study examined the effects of pan-frying, deep-frying, baking, and boiling on the amino acid and fatty acid contents, as well as the flavor, edible and nutritional qualities of BTS meat through correlation analyses.

## 2. Results

### 2.1. Sensory Evaluation

Figure 1 and Table 1 show the sensory evaluation results of BTS meat processed by different heat treatments. The aroma score of pan-fried and deep-fried meat was higher than that of baked and boiled meat. Pan-fried BTS meat had the highest aroma score (Table 1) with dominant pleasant aroma attributes such as fat-like and umami aromas. Concerning meat texture, pan-fried and baked meat scored significantly higher than deep-fried and boiled meat (*p* < 0.05). Meanwhile, different hot processing methods significantly affected the color and appearance of BTS meat. The best color was of pan-fried meat (*p* < 0.05), followed by baked meat. However, the baked BTS meat scored the highest in appearance score (*p* < 0.05). Overall, as shown in Figure 1 radar chart, pan-fried BTS meat scored had the best sensory attributes.

### 2.2. Edible Quality

Table 2 shows the difference in the edible quality of differently processed BTS meat. The shearing force of boiled and baked meat was significantly higher than that of pan-fried and deep-fried meat (*p* < 0.05); the shearing force of pan-fried BTS meat was the lowest (*p* < 0.05). The hardness of deep-fried meat was the highest (*p* < 0.05), while the other three sample groups showed no significant difference among them. Meanwhile, the chewing index of deep-fried meat was the highest, and that of pan-fried was the lowest (*p* < 0.05). In addition, the largest post-processing color difference was noticed for deep-fried meat and the lowest for boiled meat (*p* < 0.05). The elasticity and cohesion were the largest for the boiled BTS meat.

### 2.3. Nutritional Quality

Table 3 lists the effects of different cooking methods on the chemical composition of BTS meat. Deep-fried BTS meat had the lowest moisture content (58.45%) (*p* < 0.05), while the water content of pan-fried, baked, and boiled meat was more than 60%. The fat content was the highest in boiled BTS meat (*p* < 0.05), while the other three sample groups showed no significant difference among them (*p* > 0.05); nonetheless, the small difference in fat content had a trend of pan-fried meat >fried meat >baked meat. The protein content was significantly higher in pan-fried and deep-fried meat than in baked and boiled meat (*p* < 0.05); the lowest protein content was in boiled meat (*p* < 0.05).

### 2.4. Amino Acids Content

The amino acid content in differently processed BTS meat was detected by UPLC-MS technology (Table 4 and Figure 2). In total, 28 kinds of amino acids were detected. When compared with other sample groups, the amino acids (both essential and nonessential) were significantly lower in boiled BTS meat (*p* < 0.05). The maximum amino acid content was found in pan-fried meat (*p* < 0.05); meanwhile, deep-fried and baked meat had almost similar amino acid content (*p* > 0.05). Pan-frying, deep-frying, and baking treatments did not significantly affect the content of essential amino acids (*p* > 0.05). Cooking at high temperatures improved the contents of alanine, glycine, glutamine, creatine, ornithine, taurine, choline, and aminoadipic acid in BTS meat. Among these, the contents of alanine, glutamine, and ornithine were the highest in pan-fried meat (*p* < 0.05). The contents of glutamine and creatine were the lowest in deep-fried meat. Pan-frying, deep-frying, and baking significantly affected the content of ornithine in BTS meat; it was the maximum in pan-fried meat and lowest in deep-fried meat (*p* < 0.05).

As shown in Figure 2, the proportions of umami, sweet, and bitter amino acids in BTS meat samples had a consistent trend of sweet amino acids >bitter amino acids >umami amino acids. The content of umami and sweet amino acids was higher in pan-fried meat than in others.

### 2.5. Fatty Acids Content

The effect of different heat treatments on the fatty acids content of BTS meat is shown in Figure 3 and Table 5. In total, 49 free fatty acids, including 14 kinds of saturated fatty acids (SFCs), 21 kinds of monounsaturated fatty acids (MUFAs), and 14 kinds of polyunsaturated fatty acids (PUFAs), were separated and identified from experimental samples by GC-MS (Table 5). After high-temperature processing, the fatty acid composition of BTS meat has a trend of MUFAs >SFAs >PUFAs. There was no significant difference in SFAs content among the four kinds of heat-processed BTS meat samples (*p* > 0.05). However, the content of MUFAs and PUFAs was significantly higher in pan-fried (maximum) and deep-fried meat than in baked and boiled meat (*p* < 0.05). The main fatty acids in heat-treated BTS meat were oleic (C18:1N9C), linoleic (C18:2N6), palmitic (C16:0), and stearic (C18:0) acids. The C18:1N9C content was the highest in pan-fried meat, followed by deep-fried meat, and then in baked meat. The C18:2N6 content was significantly higher in pan-fried and deep-fried meat than in baked and boiled meat (*p* < 0.05). There was no significant difference in γ-linolenic acid content among the four cooking methods, but the content was higher in pan-fried and deep-fried meat than in baked and boiled meat. The α-linolenic acid content was the highest in pan-fried and deep-fried meat. There was no significant effect of different cooking methods on other trans fatty acids; only C19:1N9T and C20:1T were significantly higher in pan-fried and deep-fried meat than in baked and boiled meat (*p* < 0.05). Finally, the PUFA/SFA and n-6/n-3 values were significantly higher in pan-fried and deep-fried meat than in baked and boiled meat (*p* < 0.05).

Based on the above results of sensory evaluation, edible and nutritional quality, and amino acids and fatty acids contents, the analysis of the volatile components was mainly compared for pan-frying, deep-frying, and baking, excluding boiling.

### 2.6. Flavour

#### 2.6.1. Topographic Plots of VFCs

HS-GC-IMS technology was used to detect VFCs in the experimental samples, and the results are visually expressed through the three-dimensional topographic map (Figure 4). X, Y, and Z axes respectively represent the migration time, retention time, and peak intensity. As shown in Figure 4, differently processed BTS meat samples had similar VFCs, but their peak intensities varied depending on the heat treatment method.

Figure 5 is a top-view projection of the three-dimensional spectrum shown in Figure 4 on a two-dimensional plane. The whole spectrum represents all the VFCs in the three kinds of heat-treated meat samples. The red vertical line at abscissa 1.0 in Figure 5a is the reaction ion peak after normalization. Each different color spot represents the concentration of the respective VFC; red and white represent high and low concentrations, respectively. The drift time range of most volatile flavor substances in the sample was 1.0–2.0 ms, and the retention time was 100–800 s. Figure 5b shows the difference in samples observed by the difference comparison mode. The spectrogram of pan-fried BTS meat was selected as the reference, and the spectrograms of deep-fried and baked BTS meat were deduced by subtracting the reference. White indicates that the concentration of the volatile organic compound between the two samples is consistent, red indicates that the concentration of the substance is higher than the reference, and blue indicates that the concentration of the substance is lower than the reference. There were more blue spots in the baked BTS meat group, indicating a lower concentration of flavor compounds in baked BTS meat.

#### 2.6.2. Qualitative Analysis of VFCs

HS-GC-IMS can quickly detect volatile flavor components by headspace sampling without complex sample pretreatment, and the same was used for VFCs analysis in differently processed BTS meat samples (Table 6). The edible oil used for pan-frying and deep-frying is soybean vegetable oil. The study found that soybean oil has fewer kinds of volatile substances and a lower content of them, in which the content of hexanal, pentanal, (E) - 2-heptanal, (E) - 2-decenal and nonanal is higher. In total, 74 peaks associated with 56 VFCs (Figure 6 and Table 6) were detected, including 11 aldehydes, 10 esters, 9 alcohols, 6 ketones, and 4 pyrazines. In addition, organic acid, furan, hydroxy ketone, and sulfide were also detected. Since some compounds may have high proton affinity, they formed dimers or trimers during migration [15].

The proportion of different major compounds in the total VFCS is shown in Figure 7. Aldehydes, ketones, and alcohols were the main volatile flavor substances in BTS meat cooked at high temperatures. Aldehydes and ketones were the highest in pan-fried meat, alcohols and pyrazines were the highest in deep-fried meat, and pyrazines were absent in baked meat. As shown in Figure 6 and Table 6, the distribution of volatile flavor substances changed in different meat samples. There was no significant difference in the content of VFCs in region E among the three kinds of meat samples. The content of VFCs in region A and region F (especially 1-octene-3-one, (E)-2-octenal, 5-methyl-2-furanol, 1-pentanol, and 2-hexene-1-ol acetate) was relatively high in pan-fried meat. The content of VFCs in region B (including 1-heptanol, methyl acetate, 2-heptanone monomer, 3-methylbutyraldehyde, and valeraldehyde) was high in pan-fried and deep-fried BTS meat. Region C VFCs (n-butanol dimer, 2,5-dimethylpyrazine monomer, and 2-ethyl-3-methylpyrazine) were the highest in deep-fried meat. Many region D VFCs (including methyl hexanoate, 2-ethylfuran, 3-hydroxy-2-butanone, n-nonaldehyde dimer, hexanal dimer, and heptaldehyde monomer) were prominent in baked meat.

#### 2.6.3. OPLS-DA Analysis

To further explore the difference of VFCs in differently cooked BTS meat, the VFC data were examined by a chemometric method based on OPLS-DA. As shown in Figure 8a, the cumulative statistics R^2^X = 0.945, the model interpretation rate parameter R^2^Y = 0.986, and the prediction ability parameter Q^2^ = 0.972, all are > 0.5, indicating that the OPLS-DA model has a good prediction ability for the analysis of VFCs in different meat samples. The three kinds of meat samples clustered well on the OPLS-DA score scatter chart, with small differences within the group, while the samples from different groups were completely separated.

In addition, the OPLS-DA model was verified as shown in Figure 8b. The abscissa in the figure represents the sample retention during the displacement test. The point where the sample retention is equal to 1.0 is R^2^ and Q^2^ was obtained from the original OPLS-DA model. In the displacement test, if all R^2^ and Q^2^ values are lower than the value reserved by displacement equal to 1.0, and the regression line of the Q^2^ point crosses the abscissa or is less than zero, it is generally considered that the intercept is negative, the statistical model is valid, and there is no overfitting. After 200 cross-verifications, the Q^2^ regression line of the model still crossed the abscissa, and the intercept of the cross with the ordinate was less than zero, indicating that the model is reliable without overfitting.

The VIP value of a volatile flavor substance reflects its contribution to the model classification, and VIP > 1 is commonly used for screening different volatile flavor substances. The VIP values of different volatile flavor substances are shown in Figure 8c. In total, twenty VFCs have VIP > 1, including seven aldehydes, seven ketones, three alcohols, two esters, and one furan. The contents of 2-octenal (E), n-nonanal M, n-nonanal T, octanal M, octanal D, and heptanal M were higher in deep-fried BTS meat, while benzaldehyde D was more in baked meat.

### 2.7. Correlation Analysis of Fatty Acids and Flavor

Next, we performed a correlation analysis of fatty acids with volatile flavor substances in differently processed BTS meat samples. The fatty acids and VFCs with significant differences in the three groups (pan-fried, deep-fried, and baked) of meat samples were selected for Pearson correlation analysis (Figure 9). We found that the content of most volatile flavor compounds was significantly positively correlated with the content of fatty acids (*p* < 0.05), while benzaldehyde D, methyl hexanoate, isobutyl isovalerate, and n-Hexanol were significantly negatively correlated (*p* < 0.05). Additionally, most volatile flavor compounds were positively correlated with MUFA and PUFA (*p* < 0.05). The compounds 3-methylbutanal and 2-heptanone were positively correlated with most unsaturated fatty acids (UFAs) (*p* < 0.05), such as eicosenoic, oleic, isooleic, linoleic, α- Linolenic, and eicosapentaenoic acids. In addition, among MUFAs, isooleic acid and diethyl malonate showed a very significant positive correlation (*p* < 0.01); Among PUFAs, linoleic acid was positively correlated with methyl acetate, diethyl malonate, and ethylene glycol monoether (*p* < 0.05).

## 3. Discussion

### 3.1. Effect of Different Processing Methods on the Sensory of Property of BTS Meat

High-temperature cooking changes the flavor, appearance, texture, and color of meat products through physical and chemical reactions. Our research found that pan-fried BTS meat scored the highest in flavor evaluation; boiled BTS meat scored the lowest. This is consistent with the results of fatty acids and VFCs analyses. Accordingly, pan-fried BTS meat had more and many flavor substances than boiled meat. On the other hand, compared with boiled and deep-fried meat, consumers prefer the texture of pan-fried and baked meat. Pan-frying and baking dry the meat’s surface, forming a shell, which reduces water loss and increases the juiciness of meat [16]. Therefore, pan-fried and baked meat samples are crisp outside and tender inside, with moderate hardness and juiciness. Rendón et al. [17] found that meat products having golden yellow or baking color are more attractive to consumers. On the contrary, dry, pale meat products are rejected. In this study, pan-fried, deep-fried, and baked meat had better color scores than boiled meat. Similarly, Naeem et al. [18] reported that baked, barbecued, fried, or microwave-processed meat had better color and flavor than boiled meat.

Overall, consistent with Choi et al. [19], we found that pan-fried and baked BTS meat scored best in sensory evaluation tests as well.

### 3.2. Effect of Different Processing Methods on the Edible Quality of BTS Meat

The hardness value reflects the internal binding force of meat that maintains its integrity. Masticatory force is the chewing energy required for solid food, which reflects the food’s hardness from the side. During meat cooking, temperature rises accelerate protein oxidation, which changes muscle structure, inducing water loss and protein aggregation, thus affecting the hardness of meat [20]. Deep-fried BTS meat had a large contact area and a long contact time with high-temperature vegetable oil, causing serious water loss. Therefore, the hardness and chewing power of deep-fried BTS meat were significantly higher than the other meat samples.

Shearing force is used to evaluate the tenderness of meat products, which is often affected by high-temperature processing. Meat shearing force depends on the structure and biochemical characteristics of myofibrils and connective tissue. At 50–60 °C, myofibrils contract, thus weakening the connective tissue. The degeneration of intramuscular connective tissue increases meat tenderness. In the meantime, the structural change in myofibrils increases meat toughness [21,22]. The connective tissue transformation and myofibrillar protein denaturation of meat are affected by cooking methods, time, temperature, and meat types [22,23]. In this study, the shearing force of pan-fried BTS meat was the lowest and can be attributed to the rapid increase in heat in a shorter time during pan-frying, increasing meat tenderness [24]. The highest shearing force was of boiled meat due to the longest cooking time (60 min).

### 3.3. Effect of Different Processing Methods on the Nutritional Quality of BTS Meat

The water content in meat products reflects their tenderness, juiciness, and palatability [25]. The moisture content of meat often reduces after cooking [26,27]. Goluch et al. [27] reported that water bath cooking, grilling, oven negotiation, roasting, or pan-frying significantly lowered the water content of goose breast meat. During heat processing, the decrease in meat moisture content results from water evaporation, and the contraction of muscle fibers reduces the water-holding capacity of meat [28]. Studies found that the decrease in water content of cooked meat accompanies the increase in protein and fat content [26,29]. YU et al. found that heat treatment causes a series of changes in meat proteins, including structural (denaturation of sarcoplasmic protein and myofibrillar protein) and molecular (protein carbonylation, modification of aromatic residues, generation of Maillard reaction products) changes [30], which affect the moisture, juiciness, color, etc., of meat. Previous studies indicated that the increase in fat content of meat samples after cooking can be attributed to cooking oil usage, such as in pan-frying and deep-frying [31]. It may also vary between raw materials. For example, the fat content in the head and tail end of the same longissimus dorsi muscle was found to be higher than that in the middle [32]. In this study, the water content of pan-fried and deep-fried BTS meat was lower, and the corresponding protein content was higher. The fat content of boiled meat was the highest, while pan-fried, deep-fried and baked meat showed no difference.

### 3.4. Effect of Different Processing Methods on Amino Acids Content of BTS Meat

The content and composition of amino acids play an important role in meat quality, providing nutritional value and flavor. Essential amino acids can only be obtained from food and help maintain the nitrogen balance of the body. Different cooking methods have different effects on amino acid content. Lopes et al. [33] found that the retention rate of TAA in grilling beef was higher than that in microwave, and boiling cooked beef. Wilkinson et al. [34] showed that the longest muscle sample of pork back boiled for 90 min at 60–75 °C had an amino acid retention rate of lower than 90%; however, the retention rate increased at a lower cooking temperature. Nyam et al. [35] found that the longer the chicken breast was heat treated at the same temperature, the more amino acids were lost. In this study, the amino acid content of cooked BTS meat was significantly reduced, and the amino acid content of pan-fried BTS meat was the highest. This may be because the protein denaturation of BTS meat at a higher temperature (226–228 °C) increases the release of amino acids, and the increase of water loss leads to the increase of amino acids retention rate [33]. However, the boiling time is long (60 min), which leads to the loss of amino acids in water. The loss of amino acids in cooked meat may vary depending on the meat’s amino acids composition. Lysine is the most sensitive amino acid, which easily reacts with reducing sugar, and different Maillard reaction products are formed during heating [36]. The heating of threonine breaks the disulfide bond, and free sulfide ions react to form other compounds [37]. In this study, the retention rate of creatine was the highest followed by glutamine, alanine, taurine, choline, and glycine in cooked BTS meat, but the contents of lysine, threonine and other amino acids decreased. The amino acids content was the lowest in boiled BTS meat.

Amino acids can affect meat flavor characteristics, including taste and aroma. Amino acids can be divided into bitter amino acids including tyrosine, arginine, histidine, valine, methionine, isoleucine, leucine, tryptophan, and phenylalanine; sweet amino acids include glycine, alanine, serine, threonine, proline, and lysine; umami amino acids include glutamic acid and aspartic acid [38]. It is found that aspartic acid, glycine, and alanine can form flavoring substances with soluble reducing sugar [39]. Our results showed that the contents of umami and sweet amino acids were higher in pan-fried meat than in deep-fried, baked, and boiled meat; the glycine and alanine contents were the highest in pan-fried meat.

### 3.5. Effect of Different Processing Methods on Fatty Acids Content of BTS Meat

The composition and content of fatty acids are closely related to meat quality and flavor. Additionally, dietary fatty acids are closely related to cardiovascular health; the high content of SFAs in meat products is known to affect cholesterol metabolism [40]. Oleic acid can prevent cardiovascular and cerebrovascular diseases by maintaining a low level of low-density lipoprotein and a high level of high-density lipoprotein [41]. The amount of oleic acid synthesized by the human body is not enough and therefore must be obtained from food. α-linolenic acid is an important precursor of n-3 PUFA and α- linolenic acid (essential fatty acid) that has neuroprotective and anti-aging functions [42]. Different cooking methods have different effects on food fatty acids. Haak et al. [43] found that the content of SFAs and UFAs in oils rich in unsaturated fatty acids would be significantly reduced after cooking. The previous research of the research group found that [44] BTS meat is a high-quality raw material with UFA content being significantly higher than SFA content. Ge et al. [45] found that MUFAs and PUFAs in pan-fried and fried beef and pork are higher than those in boiled and roasted beef and pork. In this study, the content of MUFAs and PUFAs in pan-fried and deep-fried BTS meat was significantly higher than that in baked and boiled BTS meat; this was consistent with the research results of Ge et al. [45]. PUFAs are mainly polar lipids. Water loss, lipid oxidation, diffusion, and exchange during cooking reduce PUFAs content [46]. Naeem et al. [18] found that MUFAs, PUFAs, and total fatty acids in boiled rabbit meat were significantly higher than those in oven-baked, gridding baked, pan-fried, or microwave-cooked meat, but the SFAs decreased. Yu et al. [47] showed that fried chicken had the highest content of linoleic, eicosapentaenoic, and docosahexaenoic acids, roasted chicken had the highest content of oleic acid, and boiled chicken had the highest content of arachidonic acid. Perhaps due to the difference in raw materials, cooking temperature, and time, our findings are contrary to them. The SFAs, MUFAs, and PUFAs were significantly lower in boiled BTS meat than in pan-fried, deep-fried, and baked meat. The contents of linoleic and oleic acids were higher in pan-fried and deep-fried BTS meat than in baked and boiled meat, while the contents of eicosapentaenoic, docosahexaenoic, and arachidonic acids showed no significant difference.

PUFA/SFA and n-6/n-3 ratios are the indicators of fat nutritional value [48]. High PUFA/SFA ratios are believed to reduce the risk of cholesterol, while a high n-6/n-3 ratio can cause cardiovascular diseases [49]. The UK Ministry of Health suggested that the PUFA/SFA ratio in the human diet should be >0.45 [50]. The WTO/FAO recommends a healthy n-6: n-3 ratio in the range of 2.5–8:1. Another study showed that a 10:1 ratio of n-6/n-3 can decrease the risk of cardiovascular diseases, obesity, and other chronic diseases [51]. In this study, the PUFA/SFA ratio ranged from 0.78–0.18 for BTS meat from different cooking methods, and the proportion of n-6/n-3 was 8.93–1.13. The PUFA/SFA ratio and the n-6/n-3 ratio of pan-fried and deep-fried meat were within the recommended range.

### 3.6. Effect of Different Processing Methods on the Flavor of BTS Meat

Meat flavor is considered to be the most important factor affecting palatability. The main flavor precursors in meat include amino acids, fatty acids, thiamine, reducing sugar, etc. These substances form unique flavor substances in meat through thermal degradation, fat oxidation, and Maillard reactions during cooking, and include aldehydes, alcohols, ketones, esters, etc. [52]. Aldehydes are the main volatile flavor substances in cooked mutton, rabbit and horse meat [15,53,54]. Most aldehydes are produced by lipid oxidation. Elmore et al. [55] reported that PUFAs content in meat positively correlates with aldehyde content. Hexanal and pentanal are the products of oxidative degradation of linoleic acid, which provide fresh grass flavor and roasted nut aroma, respectively [56]. Linoleic acid can also be oxidized to form (E) - 2-heptanal and (E) -2-octenal, which have fresh cucumber and fruit flavors. Aldehydes (heptanal, octanal, and n-nonanal) derived from oleic acid also contribute to meat flavor and provide pleasant fatty, fruity and sweet flavor [57]. Benzaldehyde, which imparts almond and nut flavor, is the only aromatic aldehyde in cooked BTS meat and is produced by phenylalanine Strecker degradation or linolenic acid oxidation [58]. Our research results too showed that aldehydes are the main flavor in BTS meat after heat processing. The highest aldehyde content was in pan-fried meat along with more UFAs.

Alcohols and ketones are formed by carbohydrate metabolism, lipid oxidation, amino acid decarboxylation, and dehydrogenation [59]. Straight-chain ketones are mainly produced by lipid degradation, and hydroxy ketones are sugar degradation products of the Maillard reaction [60]. Most ketones have stable flavors of fruit and cream, and their threshold value is higher than that of aldehydes but lower than that of alcohols [58]. Some ketones are important intermediates of heterocyclic compounds [61], which play an important role in meat flavor. Therefore, ketones indirectly contribute to meat aroma. 1-octene-3-one has a typical mushroom and herb flavor with a low threshold value of 0.05 ppb [62]. 2-Heptanone is a common methyl ketone in meat products, which provides a fat aroma. It is formed from the β-oxidation of SFAs [63].

Alcohols are mainly formed by the oxidation of linoleic acid degradation products [64], and their threshold value is often high than aldehydes. However, alcohols have a synergistic effect with other flavor compounds [65]. At high concentrations, alcohol imparts vanilla, woody, and fat flavors to mutton. 1-pentanol is lipid-derived and provides a sweet and pleasant aroma to meat [63]. It is only found in cooked meat and the content may vary by cooking technology. 1-pentanol was found in cooked mutton and cooked beef, which was produced by the reduction of glutaraldehyde [66,67].

Esters in meat products are formed by the esterification of carboxylic acids and alcohols and have a relatively low odor threshold [68]. Generally, esters of short-chain fatty acids provide fruit and sweet taste, while esters of long-chain fatty acids produce a fatty smell [69]. We screened different volatile flavor compounds based on VIP score in OPLS-DA and found that the 2-hexene-1-ol acetate monomer had the largest VIP value. It was significantly higher in pan-fried BTS meat than in deep-fried and baked meat. Furan and pyrazine originate from the Maillard reaction and Strecker degradation [70]. Pyrazine has a strong aroma of roasted almond, nut, and cocoa flavor. We found that the content of pyrazine was the highest in deep-fried meat, while it was absent in baked meat. It showed that the Maillard reaction intensity was higher in deep-fried meat than in pan-fried and baked meat, which is consistent with the chromaticity results. Furan compounds are formed from a variety of sources, such as from the oxidation of α-linolenic acid, γ-linolenic acid [56], and linoleic acid [71]. Sulfur compounds are formed by the degradation of thiamine [54] and have an onion/garlic odor. Roldán et al. [15] detected furan and sulfur compounds in cooked lamb tenderloin.

### 3.7. Correlation Analysis of Fatty Acids and Flavor Substances

Heat treatment improves the flavor of meat products as the fat in the meat is decomposed into free fatty acids that form volatile flavor substances; therefore, the difference in fatty acids composition in mutton has a greater impact on its flavor [72]. In Figure 9, most flavor substances positively correlated with fatty acids, indicating that these originated from the oxidation and degradation of fatty acids. Butanal, 3-methylbutal, 3-methyl-2-butenal, methyl acetate, diethyl malonate, ethanol, 2-heptanone M, 2-heptanone D, 1-heptanol, and 2-butanone T have the flavor of cocoa, fruit, chocolate, wine, cheese, garlic, and mustard, and positively affected the flavor of BTS mutton; however, allyl isothiocyanate and furfuryl methyl sulfide have an irritating taste and negatively impacted the flavor, which produced from lipid peroxidation during high-temperature cooking [73]. We found that MUFAs and PUFAs were significantly positively correlated with most flavor substances, indicating their contribution to the formation of volatile flavor substances.

## 4. Materials and Methods

### 4.1. Materials and Sample Preparation

BTS were purchased from the Black Tibetan Sheep Breeding Center in Guinan County, Qinghai Province, China. These animals were raised from hatchlings on standardized commercial farms with consistent rearing conditions and similar diets (commercial sheep feed); the sheep were reared for 4 months (average body weight 36.42 ± 0.75 kg). Sheep were fasted for three hours before slaughter and were exsanguinated by bolt stunning. After slaughtering, the hind leg meat of BTS was sorted; the visible fat, hard bone, cartilage, lymph, and fascia were removed. After cleaning the skin surface from dirt, the hind leg meat was evenly cut into meat sticks of size 1 × 1 × 3 cm. All meat sticks were snap-frozen in liquid nitrogen before storage at −80 °C until further processing.

The meat was subjected to different heat processing treatments. For pan-frying, the meat was put with 10 ml of edible oil in the heating pot and decocting was done for 3 min at 226–228 °C, 1.5 min for each side. For deep-frying, the meat was put with 50 ml of edible oil in the heating pot and fried at 226–228 °C for 4 min with continuous stirring. For baking, the meat was wrapped in tin foil and roasted at 180 °C for 20 min, turning once, and then baked for an additional 20 min. For the boiling treatment, the meat was boiled in 2.5 L water for 60 min. All cooked meat stick samples were vacuum packed and stored at -80 °C until used for volatile compound analysis.

### 4.2. Meat Quality Indices

#### 4.2.1. Sensory Evaluation

The sensory evaluation analysis was performed based on previous literature with some modifications [74]. For the evaluation of cooked meat samples, we invited 10 food professionals (five males and five females). They were already trained in sensory evaluation and familiar with evaluating meat products. The four sets (pan-frying, deep-frying, baking, and boiling) of cooked meat samples were divided into forty portions, numbered, and marked with a random three-digit code. Five aromatic attributes (appearance, color, texture, smell, and acceptability) and their corresponding reference standards were used by the panelists to evaluate the aroma profiles of meat samples (Table 7). Each member had a separate booth in a controlled room, and the environment was kept as constant as possible. Each member received two sets of samples in a randomized order for scoring during each session. The evaluators were not allowed to communicate results with each other. The evaluators were asked to thoroughly rinse the mouth with water between two evaluations at an interval of hours. The experiment was performed in triplicates.

#### 4.2.2. Edible quality examination

The determination of meat edible quality included pH, shear force, chromaticity, and texture. The shear force (unit, N) of the meat sample (1 × 1 × 1 cm) was measured using the tenderness meter. The color value (△E) of the muscle cross-section was measured using a standardized color meter. Texture determination was performed of a 1 × 1 × 1 cm meat sample; the instrument parameters were set in the TPA mode: test probe, ta36; test fixture, ta-sba; test speed, 2.00 mm/s; repeated 3 times. The measured indexes included hardness, elasticity, cohesion, and chewing ability. All experiments were performed three times to determine the average values.

#### 4.2.3. Nutritional Quality Examination

The water, protein, and fat contents of meat samples were determined following the specified meat detection standards [75]. The specific measurement was based on the recognized AOAC methods. Briefly, moisture measurement was estimated by oven-drying the sample at 105 °C until the sample reached a constant weight. The Kjeldahl method was used for protein determination; the nitrogen-to-protein conversion coefficient was 6.25. The Soxhlet extractor was used for fat determination.

#### 4.2.4. Amino Acids Examination

A total of 50 mg of the stored (at −80 °C) meat sample was transferred to a 2 ml test tube and added with 600 µL of 10% formic acid methanol solution with 60 s of vortexing. Then, two low-temperature ultrasonication treatments of 30 min each were performed. Afterward, the samples were kept at −20 °C for an hour to precipitate protein, followed by a 20-min centrifugation at 4 °C and 14,000 rcf. The collected supernatant was diluted appropriately, of which, 100 µL was added with 100 µL double isotope internal standard (100 ppb) by vortexing for 30 s. This mixed sample was injected into the detection system. An ultra-performance liquid chromatography (UPLC) system (Agilent 1290 Infinity LC) equipped with a 5500 qtrap mass spectrometer was used for amino acid determination. A quality control sample (QC) was used to assess the repeatability and stability of the system. The detection conditions were as follows: column temperature 40 °C, flow rate 250 μL/min, injection volume 1 μL, and mobile phase: liquid A = 25 mM ammonium formate +0.08% FA aqueous solution, liquid B = 0.1% FA acetonitrile. Gradient elution conditions were as follows: 0–12 min, 90–70% B solution; 12–18 min, 70–50% B solution; 18–25 min, 50–40% B solution; 30–30.1 min, 40–90% B solution. The mass spectrometry was performed in positive ion mode with an electrospray spray ionization source in the multi-reaction monitoring scanning mode; the ion power temperature was 500 °C, the ion source voltage was 5500 v, the pressure of ion source gas 1 and 2 was 40 psi, and the pressure of curtain gas was 30 psi.

#### 4.2.5. Fatty Acids Determination

Fatty acid methyl esters standard solutions were prepared in n-hexane at 5, 10, 25, 50, 100, 250, 500, 1000, and 2000 µg/mL. Then, 50 mg of meat sample was transferred to a 2 ml glass centrifuge tube and added with 1 ml of chloroform–methanol (2:1) solution. The mixture was ultrasonicated for 30 min and then centrifuged. The collected supernatant was added with 2 ml of 1% sulfuric acid–methanol solution and incubated at 80 °C for 0.5 h. The mixture was then added with 1 ml of n-hexane for extraction and finally washed with 5 ml of pure water. After centrifugation, the supernatant was mixed well with 15 uL of 500 ppm methyl salicylate (internal standard). Then, 200 uL of supernatant was absorbed into the detection bottle. The Thermo trace 1300/tsq 9000 gas mass spectrometer (GC-MS) was used for fatty acid determination, and the Thermo tg-fame capillary column was used for gas chromatography (GC): the injection volume was 1 uL, the split ratio was 8:1, the sample inlet temperature was 250 °C, the ion source temperature was 300 °C, and the temperature of the transmission line was 280 °C. The MS conditions were as follows: injection port temperature, 280 °C; ion source temperature, 230 °C; transmission line temperature, 250 °C; ionization, electron bombardment ionization (EI) source; scanning mode, SIM; electron energy, 70 ev.

#### 4.2.6. Volatile Flavor Compounds Determined by HS-GC-IMS

The VFCs in meat samples were analyzed by Headspace-gas chromatography-ion mobility spectrometry (HS-GC-IMS) FlavourSpec® flavor analyzer, (the G.A.S. Department of Shandong Hai Neng Science Instrument Co., Ltd., Shan-dong, China). The determination of VFCs was performed as described by Xuan et al. [76] with slight modifications. Briefly, 2 g of minced meat sample was placed in the 20 mL headspace sample bottle for GC-IMS analysis. The sample bottle was incubated for 15 min at 80 °C. The incubation speed was 500 rpm; the headspace gas phase (500 μL) was automatically injected at 85 °C in splitless injection mode. GC equipped with an MXT-5 (15 m x 0.53 mm) column was used to separate the mixture at 60 °C. The carrier gas was N2 (purity ≥ 99.99 %), and the instrument running time was 20 min. The flow program was as follows: the initial flow rate of 2 mL/min was maintained for 2 min to separate difficult substances. Then, the flow rate was increased to 10 mL/min within 8 min; next, the flow was linearly ramped up to 150 mL/min within 20 min; after that, the analysis was stopped. The analyte was separated at 60 °C by a column and then ionized in a positive ion mode in an IMS ionization chamber having a 6.5 KeV tritium ionization source. Purified N2 (purity ≥ 99.99%) was used as drift gas in IMS at a flow rate of 150 mL/min, and the drift tube temperature was constant at 45 °C. All analyses were performed in triplicates.

#### 4.2.7. Statistical Analysis

All data were first sorted by Excel software and then analyzed by SPSS 25.0 software using ANOVA and Duncan’s method for multiple comparisons between groups. The final results are expressed as “mean ± standard deviation”. Data with *p* < 0.05 indicate a significant difference. SPSS 25.0 was used for Pearson correlation analysis. OriginPro 2021 and SPSS 25.0 were used to draw radar charts, histograms, and correlation heat maps. NIST and IMS databases built in the VOCal software were used for qualitative analysis of substances; the Reporter plug-in was used to directly compare the spectral differences between samples; the Gallery Plot plug-in was used to compare fingerprints. SIMCA 14.1 was used to establish orthogonal partial least squares discriminant analysis model (OPLS-DA).

## 5. Summary

The present investigation focused on the effect of different heat processing methods on the quality and flavor of BTS meat. The meat quality analysis showed that pan-fried BTS meat has the highest overall acceptability, lowest shearing force, and highest protein, essential amino acids, and non-essential amino acids content. From the flavor standpoint, in total, 56 VFCs were identified in processed BTS meat by HS-GC-IMS; esters, aldehydes, ketones, and alcohols were the main flavor compounds, among which aldehydes contributed the most to the overall flavor of BTS meat. Pan-fried BTS meat could retain most flavors compared with deep-fried or baked meat. The key flavor compounds 2-hexen-1-ol, 1-octen-3-one, 1-pentanol, 2-octenal, octanal, heptanal, and benzaldehyde were summed up, as identified by OPLS-DA and VIP values. Fatty acids (C18:1N9C, C18:1N7, C18:2N6, and C18:3N3) crucially contributed to the characteristic flavor of pan-fried BTS meat. The specific mechanism of flavor changes in BTS meat during different heat processing methods will be explored in the future.

## Figures and Tables

**Figure 1 molecules-28-00165-f001:**
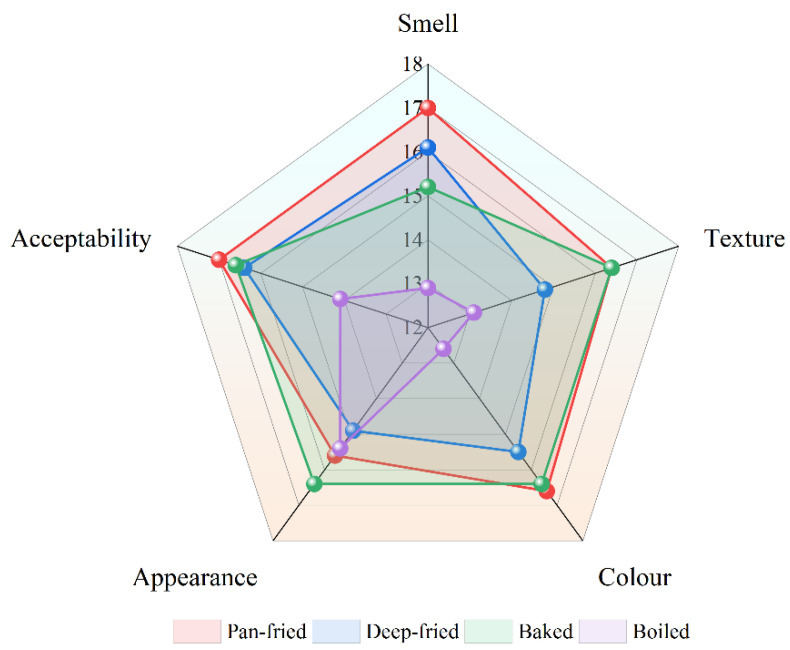
Radar Chart of Sensory Score of BTS meat under Different Processing Methods.

**Figure 2 molecules-28-00165-f002:**
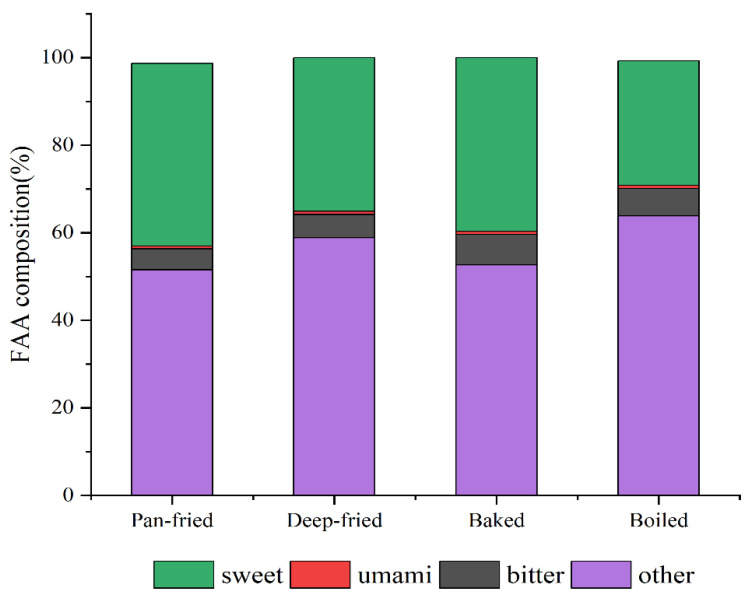
Amino acids composition of BTS meat after different thermal processing methods; sweet = (glycine + alanine + serine + threonine + proline + lysine); umami = (glutamic + acidaspartic acid); bitter = (tyrosine + arginine + histidine + valine + methionine + isoleucine + leucine + tryptophan + phenylalanine).

**Figure 3 molecules-28-00165-f003:**
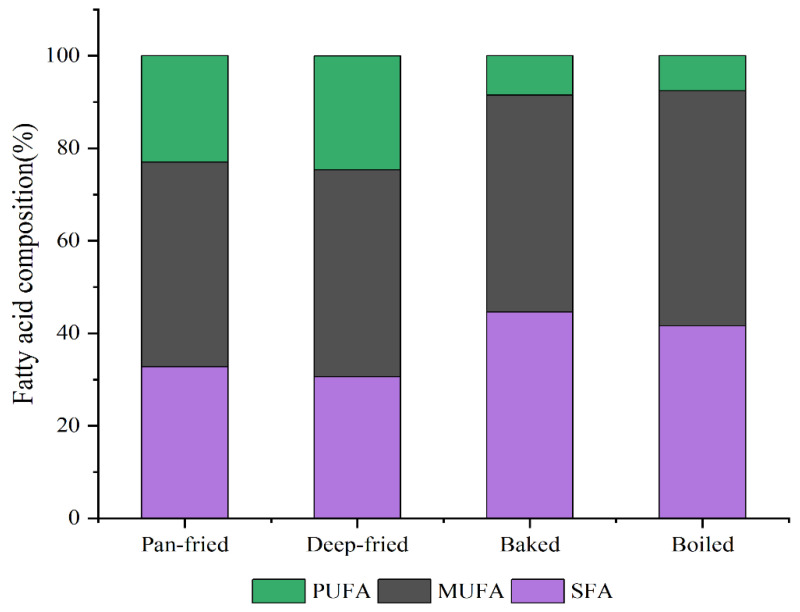
Fatty acid composition of BTS meat after different thermal processing; PUFA: polyunsaturated fatty acids; MUFA: monounsaturated fatty acids; SFA: saturated fatty acids.

**Figure 4 molecules-28-00165-f004:**
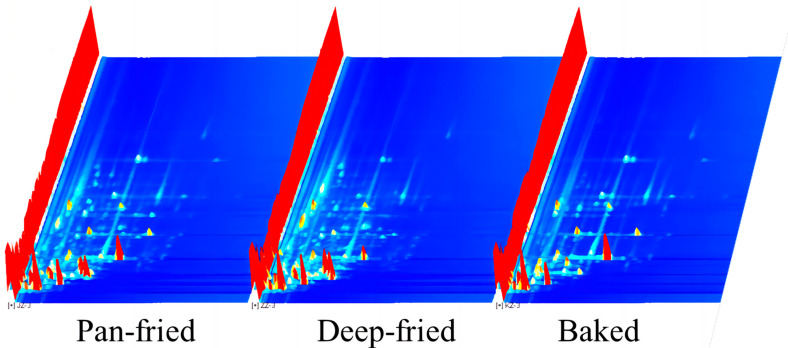
3D topographic map of BTS meat processed by different high-temperature cooking methods.

**Figure 5 molecules-28-00165-f005:**
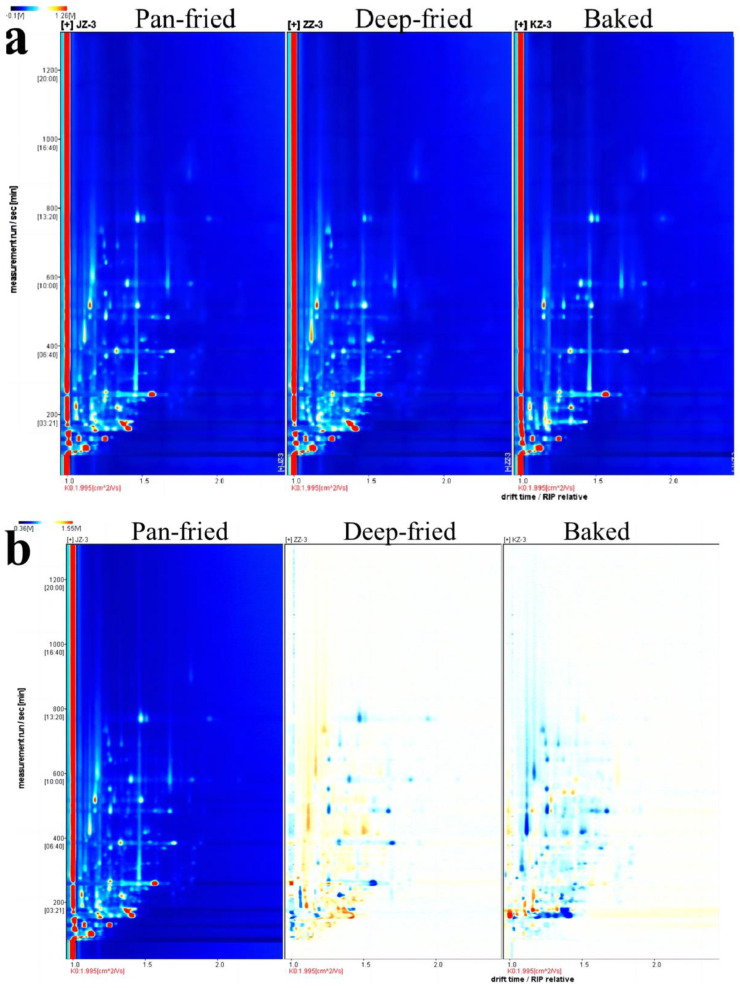
2D topographic map of BTS meat processed by different high-temperature cooking methods. (**a**): ion mobility spectrogram; (**b**): comparison results under the spectral diagram of pan-fried sample was selected as the reference.

**Figure 6 molecules-28-00165-f006:**
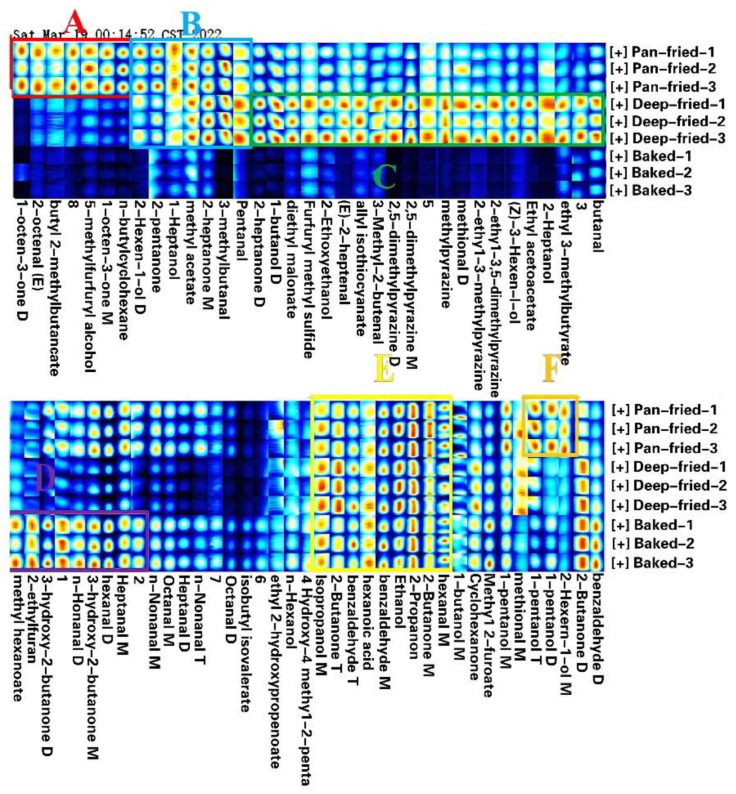
Fingerprints of flavor substances in BTS meat processed by different methods. (**A**,**F**): VFCs with higher content in pan-fried; (**B**): VFCs with higher content in pan-fried and deep-fried; (**C**): VFCs with higher content in deep-fried; (**D**): VFCs with higher content in baked; (**E**): VFCs with higher content in pan-fried, deep-fried and baked.

**Figure 7 molecules-28-00165-f007:**
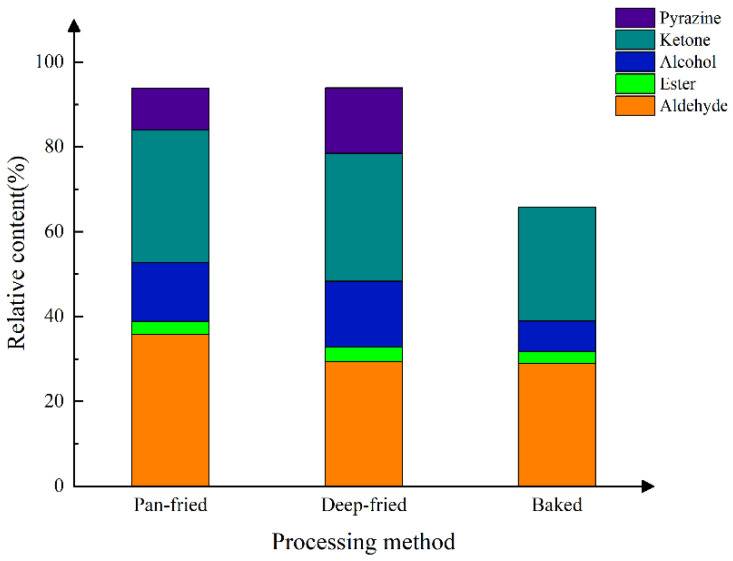
Relative contents of various flavor substances in BTS meat processed by different methods.

**Figure 8 molecules-28-00165-f008:**
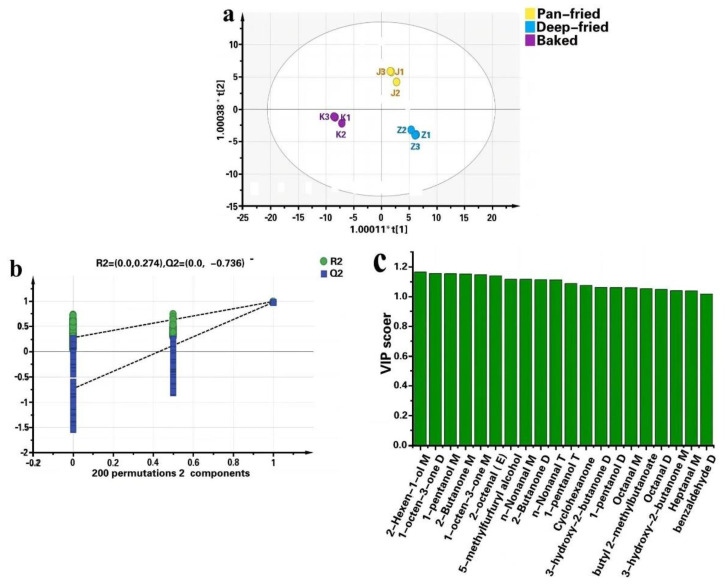
OPLS-DA score plot of volatile flavor compounds in pan-fried, deep- fried and baked BTS meat. (**a**): score plot; (**b**): permutation test plot; (**c**): VIP(variable importance for predictive components) values of VFCs.

**Figure 9 molecules-28-00165-f009:**
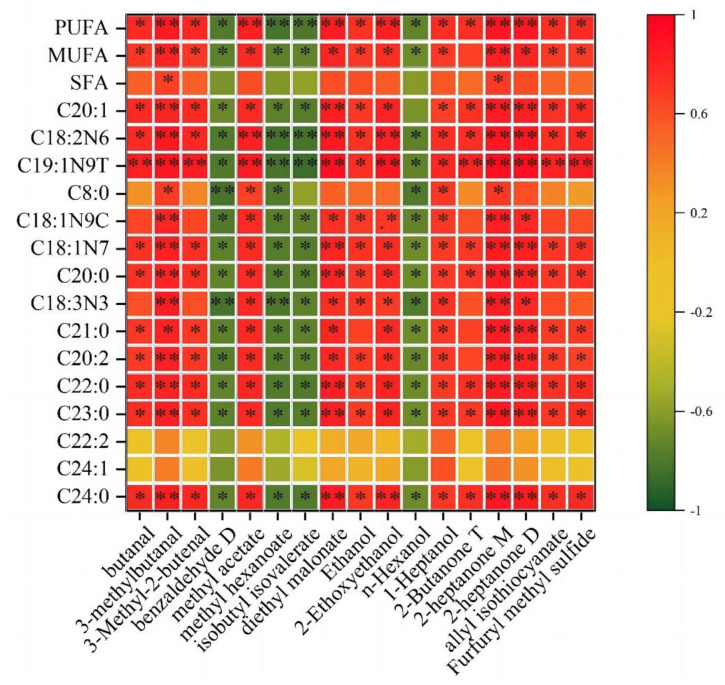
Correlation heat map between fatty acids and volatile flavor substances in heat processing BTS meat.

**Table 1 molecules-28-00165-t001:** Analysis of Sensory Assessment Data of BTS meat under Different Processing Methods.

	Evaluation Index				
Processing	Smell	Texture	Colour	Appearance	Acceptability
Pan-fried	17.00 ± 0.42 ^a^	16.40 ± 0.40 ^a^	16.60 ± 0.37 ^a^	15.60 ± 0.31 ^ab^	17.00 ± 0.26 ^a^
Deep-fried	16.10 ± 0.41 ^ab^	14.80 ± 0.36 ^b^	15.50 ± 0.40 ^b^	14.90 ± 0.38 ^b^	16.40 ± 0.34 ^a^
Baked	15.20 ± 0.33 ^b^	16.40 ± 0.34 ^a^	16.40 ± 0.31 ^ab^	16.40 ± 0.27 ^a^	16.60 ± 00.27 ^a^
Boiled	12.90 ± 0.38 ^c^	13.10 ± 0.31 ^c^	12.60 ± 0.27 ^c^	15.40 ± 0.40 ^ab^	14.10 ± 0.38 ^b^

The data in the same column marked with different letters showed significant differences (*p* < 0.05), while the data marked with the same letter or no letter showed no significant differences (*p* > 0.05).

**Table 2 molecules-28-00165-t002:** Edible quality indicators of differently processed BTS meat.

Processing	Shearing Force (N)	Chroma (△E)	Hardness (g)	Elastic (mm)	Cohesion	Chewing Ability (mJ)
Boiled	4.70 ± 0.14 ^a^	77.91 ± 0.18 ^d^	1431.00 ± 172.40 ^b^	3.72 ± 0.42 ^a^	0.73 ± 0.08 ^a^	34.45 ± 3.37 ^ab^
Pan-fried	2.96 ± 0.18 ^c^	81.79 ± 0.24 ^a^	1580.67 ± 251.22 ^b^	2.97 ± 0.43 ^ab^	0.62 ± 0.08 ^ab^	28.17 ± 6.55 ^b^
Deep-fried	3.77 ± 0.20 ^b^	82.88 ± 0.34 ^b^	2625.33 ± 100.09 ^a^	2.86 ± 0.33 ^b^	0.61 ± 0.04 ^ab^	44.33 ± 3.45 ^a^
Baked	4.45 ± 0.09 ^a^	80.73 ± 0.27 ^c^	1546.67 ± 141.74 ^b^	3.36 ± 0.16 ^ab^	0.68 ± 0.04 ^ab^	34.60 ± 5.83 ^ab^

The data in the same column marked with different letters showed significant differences (*p* < 0.05), while the data marked with the same letter or no letter showed no significant differences (*p* > 0.05).

**Table 3 molecules-28-00165-t003:** Nutritional quality indicators of differently processed BTS meat.

Processing	Water (%)	Fat (%)	Protein (%)
Boiled	65.93 ± 1.57 ^a^	6.99 ± 0.51 ^a^	32.18 ± 1.60 ^c^
Pan-fried	60.48 ± 1.12 ^b^	5.61 ± 0.35 ^b^	40.47 ± 1.11 ^a^
Deep-fried	58.45 ± 2.10 ^bc^	5.25 ± 0.10 ^b^	42.64 ± 0.20 ^a^
Baked	62.80 ± 0.93 ^ab^	4.87 ± 0.01 ^b^	37.10 ± 0.21 ^b^

The data in the same column marked with different letters showed significant differences (*p* < 0.05), while the data marked with the same letter or no letter showed no significant differences (*p* > 0.05).

**Table 4 molecules-28-00165-t004:** Amino acids of BTS meat under different processing methods umol/g.

Iterms	Pan-Fried	Deep-Fried	Baked	Boiled
Glutamate	0.25 ± 0.01 ^a^	0.15 ± 0.01 ^b^	0.26 ± 0.05 ^a^	0.03 ± 0.00 ^c^
Glycine	2.59 ± 0.05 ^a^	1.61 ± 0.12 ^b^	2.30 ± 0.46 ^ab^	0.08 ± 0.01 ^c^
Lysine	0.42 ± 0.02 ^a^	0.30 ± 0.03 ^a^	0.40 ± 0.07 ^a^	0.02 ± 0.01 ^b^
Aspartate	0.14 ± 0.03 ^ab^	0.19 ± 0.06 ^a^	0.04 ± 0.01 ^bc^	0.00 ± 0.00 ^c^
Arginine	0.51 ± 0.01 ^a^	0.39 ± 0.03 ^a^	0.43 ± 0.08 ^a^	0.03 ± 0.01 ^b^
Serine	0.64 ± 0.03 ^a^	0.50 ± 0.04 ^a^	0.58 ± 0.11 ^a^	0.03 ± 0.01 ^b^
Methionine	0.02 ± 0.00 ^b^	0.02 ± 0.00 ^b^	0.03 ± 0.00 ^a^	0.00 ± 0.00 ^c^
Phenylalanine	0.26 ± 0.02 ^a^	0.20 ± 0.03 ^a^	0.28 ± 0.05 ^a^	0.02 ± 0.01 ^b^
Tyrosine	0.21 ± 0.02 ^a^	0.18 ± 0.02 ^a^	0.23 ± 0.04 ^a^	0.02 ± 0.01 ^b^
Leucine	0.50 ± 0.03 ^a^	0.41 ± 0.06 ^a^	0.59 ± 0.12 ^a^	0.02 ± 0.01 ^b^
Isoleucine	0.32 ± 0.02 ^a^	0.26 ± 0.03 ^a^	0.36 ± 0.07 ^a^	0.02 ± 0.01 ^b^
Histidine	0.45 ± 0.04 ^ab^	0.36 ± 0.03 ^b^	0.55 ± 0.08 ^a^	0.07 ± 0.02 ^c^
Proline	0.66 ± 0.03 ^a^	0.48 ± 0.00 ^a^	0.48 ± 0.11 ^a^	0.01 ± 0.01 ^b^
Valine	0.46 ± 0.02 ^a^	0.37 ± 0.04 ^a^	0.46 ± 0.08 ^a^	0.02 ± 0.01 ^b^
Threonine	0.35 ± 0.02 ^a^	0.30 ± 0.02 ^a^	0.36 ± 0.07 ^a^	0.02 ± 0.01 ^b^
Alanine	9.14 ± 0.12 ^a^	5.96 ± 0.48 ^b^	6.68 ± 1.19 ^b^	0.39 ± 0.10 ^c^
Asparagine	0.15 ± 0.01 ^a^	0.13 ± 0.01 ^a^	0.16 ± 0.02 ^a^	0.02 ± 0.00 ^b^
Creatine	12.83 ± 0.38 ^a^	8.62 ± 0.98 ^b^	12.56 ± 1.74 ^a^	1.59 ± 0.25 ^c^
Citrulline	0.06 ± 0.00 ^a^	0.04 ± 0.00 ^a^	0.05 ± 0.02 ^a^	0.00 ± 0.00 ^b^
Glutamine	13.30 ± 0.98 ^a^	6.80 ± 0.71 ^b^	7.29 ± 1.07 ^b^	0.63 ± 0.18 ^c^
Creatinine	0.85 ± 0.06 ^a^	0.81 ± 0.02 ^a^	0.61 ± 0.10 ^b^	0.17 ± 0.02 ^c^
Tryptophan	0.28 ± 0.02 ^a^	0.21 ± 0.01 ^a^	0.24 ± 0.03 ^a^	0.06 ± 0.01 ^b^
Hydroxyproline	0.10 ± 0.00 ^a^	0.07 ± 0.00 ^b^	0.07 ± 0.01 ^b^	0.00 ± 0.00 ^c^
Ornithine	2.04 ± 0.03 ^a^	0.76 ± 0.12 ^c^	1.15 ± 0.11 ^b^	0.09 ± 0.02 ^d^
Taurine	5.71 ± 0.34 ^a^	4.56 ± 0.53 ^a^	4.89 ± 0.56 ^a^	0.58 ± 0.15 ^b^
Choline	5.06 ± 0.66 ^b^	7.59 ± 0.94 ^a^	3.17 ± 0.34 ^b^	0.18 ± 0.02 ^c^
Aminoadipic acid	5.64 ± 0.81 ^a^	4.20 ± 1.08 ^a^	1.34 ± 0.59 ^b^	0.02 ± 0.01 ^b^
EAAs/(umol/g)	2.60 ± 0.14 ^a^	2.06 ± 0.22 ^a^	2.72 ± 0.48 ^a^	0.19 ± 0.06 ^b^
NEAAs/(umol/g)	60.33 ± 1.56 ^a^	43.41 ± 3.24 ^b^	42.83 ± 5.56 ^b^	3.95 ± 0.80 ^c^
TAAs/(umol/g)	62.93 ± 1.70 ^a^	45.47 ± 3.44 ^b^	45.56 ± 6.03 ^b^	4.15 ± 0.86 ^c^

^1^ The data in the same column marked with different letters showed significant differences (*p* < 0.05), while the data marked with the same letter or no letter showed no significant differences (*p* > 0.05); ^2^ EAAs: essential amino acids, NEAAs: non-essential amino acids, TAAs: total amino acids; ^3^ EAAs = leucine + methionine + valine + isoleucine + threonine + phenylalanine + lysine + tryptophan.

**Table 5 molecules-28-00165-t005:** Fatty acids of BTS meat under different processing methods ug/g.

Fatty Acids	Pan-Fried	Deep-Fried	Baked	Boiled
C8:0	5.19 ± 1.22 ^a^	2.44 ± 0.88 ^ab^	0.11 ± 0.08 ^b^	0.33 ± 0.04 ^b^
C10:0	9.77 ± 3.21	7.12 ± 0.90	7.15 ± 0.99	8.64 ± 1.06
C12:0	8.33 ± 2.24	6.61 ± 0.69	5.90 ± 0.73	7.18 ± 0.87
C13:0	0.72 ± 0.33	0.35 ± 0.11	0.39 ± 0.03	0.49 ± 0.24
C14:0	211.02 ± 57.96	154.33 ± 19.52	134.43 ± 16.15	170.25 ± 28.08
C15:0	57.14 ± 12.85	44.22 ± 4.57	40.36 ± 5.75	53.22 ± 8.72
C16:0	4324.42 ± 710.22 ^a^	4043.59 ± 732.95 ^ab^	2287.36 ± 213.48 ^b^	2740.59 ± 352.79 ^ab^
C17:0	169.24 ± 36.80	133.46 ± 16.74	120.78 ± 14.08	152.75 ± 23.02
C18:0	2925.01 ± 499.54	2590.03 ± 397.31	2015.13 ± 177.06	2348.88 ± 293.78
C20:0	74.71 ± 12.21 ^a^	77.07 ± 22.74 ^a^	15.60 ± 2.21 ^b^	17.52 ± 1.83 ^b^
C21:0	5.01 ± 0.71 ^a^	4.89 ± 1.61 ^a^	0.69 ± 0.3 ^b^	0.80 ± 0.21 ^b^
C22:0	59.86 ± 9.85 ^a^	65.70 ± 23.07 ^a^	2.36 ± 0.76 ^b^	2.78 ± 0.31 ^b^
C23:0	6.97 ± 0.96 ^a^	7.26 ± 2.38 ^a^	0.68 ± 0.42 ^b^	0.91 ± 0.25 ^b^
C24:0	27.38 ± 4.24 ^a^	29.91 ± 9.52 ^a^	2.15 ± 0.83 ^b^	2.36 ± 0.25 ^b^
SFA	7884.77 ± 1349.43	7166.97 ± 1228.35	4632.94 ± 426.06	5506.58 ± 710.80
C14:1T	43.79 ± 6.44	37.80 ± 4.09	41.83 ± 1.47	42.28 ± 3.28
C14:1	30.93 ± 5.64	26.36 ± 2.77	30.40 ± 1.35	30.11 ± 2.02
C15:1T	32.43 ± 3.81	28.61 ± 2.22	30.10 ± 1.10	31.12 ± 1.78
C15:1	18.42 ± 0.99	17.12 ± 0.54	18.59 ± 1.14	18.62 ± 0.79
C16:1T	55.14 ± 9.53	44.78 ± 3.94	45.71 ± 3.85	55.86 ± 7.37
C16:1	157.64 ± 27.73	125.81 ± 11.60	99.45 ± 9.11	139.86 ± 26.60
C17:1T	40.25 ± 6.04	35.45 ± 3.01	38.22 ± 1.51	42.86 ± 3.13
C17:1	93.43 ± 15.55	71.70 ± 4.01	61.80 ± 6.31	69.88 ± 13.68
C18:1N12T	27.93 ± 5.08	21.68 ± 2.44	20.79 ± 2.59	27.37 ± 3.50
C18:1N9T	32.66 ± 5.14	26.48 ± 3.36	25.41 ± 3.19	38.89 ± 3.53
C18:1N7T	207.08 ± 47.22	152.89 ± 17.50	160.49 ± 21.44	213.06 ± 40.43
C18:1N12	1140.46 ± 165.32 ^b^	1344.62 ± 278.43 ^ab^	1075.66 ± 125.38 ^b^	1840.75 ± 100.95 ^a^
C18:1N9C	6183.56 ± 864.63 ^a^	5692.87 ± 1048.67 ^ab^	2652.89 ± 241.90 ^c^	3455.70 ± 588.45 ^bc^
C18:1N7	296.96 ± 49.12 ^a^	305.30 ± 81.08 ^a^	76.78 ± 6.72 ^b^	99.82 ± 16.45 ^b^
C19:1N12T	15.08 ± 4.22	11.88 ± 3.69	8.28 ± 1.82	12.33 ± 3.01
C19:1N9T	1428.81 ± 161.53 ^a^	1624.17 ± 343.70 ^a^	241.78 ± 28.81 ^b^	333.36 ± 16.61 ^b^
C20:1T	39.52 ± 5.76 ^a^	38.67 ± 10.32 ^a^	13.09 ± 0.67 ^b^	13.38 ± 0.91 ^b^
C20:1	721.69 ± 106.97 ^a^	797.2 8 ± 228.70 ^a^	169.60 ± 25.86 ^b^	204.91 ± 17.45 ^b^
C22:1N9T	14.71 ± 2.58	12.03 ± 1.22	12.42 ± 0.52	12.97 ± 0.86
C22:1N9	30.26 ± 4.33	23.64 ± 2.53	23.97 ± 1.13	24.77 ± 1.55
C24:1	33.77 ± 2.74 ^a^	25.63 ± 1.09 ^b^	24.90 ± 1.19 ^b^	25.23 ± 0.72 ^b^
MUFA	10644.52 ± 1374.73 ^a^	10464.76 ± 2035.59 ^a^	4872.17 ± 439.29 ^b^	6733.15 ± 764.07 ^b^
C18:2N6T	13.55 ± 3.35	10.50 ± 1.27	9.50 ± 1.50	13.82 ± 2.69
C18:2N6	4730.83 ± 654.90 ^a^	4998.37 ± 1441.40 ^a^	308.67 ± 54.75 ^b^	359.51 ± 38.55 ^b^
C18:3N6	4.76 ± 1.12	6.69 ± 2.98	2.22 ± 0.37	2.55 ± 0.39
C18:3N3	175.95 ± 39.56 ^a^	146.90 ± 28.92 ^a^	31.69 ± 7.87 ^b^	34.02 ± 5.01 ^b^
C20:2	14.25 ± 1.54 ^a^	13.92 ± 2.28 ^a^	7.41 ± 0.44 ^b^	8.37 ± 0.71 ^b^
C20:3N6	20.31 ± 1.66	21.31 ± 2.33	15.67 ± 1.59	17.75 ± 1.45
C20:3N3	131.90 ± 9.65	141.27 ± 7.06	123.12 ± 21.23	144.49 ± 14.80
C20:4N6	160.58 ± 16.40	153.70 ± 6.96	121.71 ± 27.18	123.71 ± 25.84
C22:2	5.64 ± 0.56 ^a^	3.94 ± 0.23 ^b^	3.96 ± 0.32 ^b^	4.03 ± 0.14 ^b^
C20:5N3	100.23 ± 9.29	104.39 ± 8.18	96.85 ± 12.37	109.60 ± 9.45
C22:4	11.50 ± 0.61	10.06 ± 0.55	9.49 ± 0.78	10.48 ± 0.99
C22:5N6	6.90 ± 0.32 ^a^	6.02 ± 0.29 ^ab^	5.54 ± 0.43 ^b^	5.96 ± 0.40 ^ab^
C22:5N3	117.09 ± 8.96	118.03 ± 5.77	106.98 ± 14.67	120.80 ± 13.75
C22:6N3	47.99 ± 3.14	50.59 ± 3.62	46.03 ± 7.69	52.00 ± 4.90
PUFAs	5527.22 ± 738.56 ^a^	5771.79 ± 1506.62 ^a^	881.43 ± 148.66 ^b^	998.73 ± 114.14 ^b^
UFAs	16171.75 ± 2103.85 ^a^	16236.54 ± 3539.91 ^a^	5753.60 ± 521.22 ^b^	7731.88 ± 811.79 ^b^
PUFAs/SFAs	0.71 ± 0.06 ^a^	0.78 ± 0.08 ^a^	0.19 ± 0.03 ^b^	0.18 ± 0.02 ^b^
n-6/n-3	8.56 ± 0.23 ^a^	8.93 ± 1.78 ^a^	1.13 ± 0.03 ^b^	1.13 ± 0.04 ^b^

^1^ The data in the same column marked with different letters showed significant differences (*p* < 0.05), while the data marked with the same letter or no letter showed no significant differences (*p* > 0.05); ^2^ SFAs: saturated fatty acids, MUFAs: monounsaturated fatty acids, PUFAs: polyunsaturated fatty acids, UFAs: Unsaturated fatty acid, n-3 PUFAs: omega-3 polyunsaturated fatty acids, n-6 PUFAs: omega-6 polyunsaturated fatty acids.

**Table 6 molecules-28-00165-t006:** Influence of different processing methods on the flavor substances of BTS meat.

Volatiles	NO.	Compounds	Retention Indexwas	Retention Times (s)	Drift Times (ms)	Odor Descriptions	Peak Volume			*p*
							Pan-Fried	Deep-Fried	Baked	
Aldehydes	1	Butanal	592.6	136.125	1.30114	Cocoa green	172.48 ± 9.20 ^b^	255.80 ± 22.93 ^a^	132.32 ± 4.81 ^c^	<0.01
	2	3-methylbutanal	654.8	161.289	1.41245	fruity chocolate	5318.01 ± 792.62 ^a^	6204.64 ± 290.44 ^a^	216.94 ± 48.09 ^b^	<0.01
	3	Pentanal	729.1	205.451	1.18641	Bready nutty	274.37 ± 9.23	285.98 ± 10.44	—	0.222
	4	3-Methyl-2-butenal	782.5	250.446	1.09537	sweet fruity	84.39 ± 3.94 ^b^	161.63 ± 19.13 ^a^	32.93 ± 6.59 ^c^	<0.01
	5	hexanal M	792.4	259.566	1.25737	fresh green	1535.10 ± 103.10	1537.42 ± 25.36	1434.85 ± 53.74	0.194
	6	hexanal D	791.7	258.989	1.56569	fresh green	4758.95 ± 630.97 ^b^	2309.95 ± 111.08 ^c^	5737.36 ± 80.22 ^a^	<0.01
	7	Heptanal M	901.8	386.032	1.32714	fresh fatty	1395.98 ± 48.07 ^a^	852.00 ± 2.20 ^b^	1488.75 ± 76.51 ^a^	<0.01
	8	Heptanal D	901.3	385.382	1.69514	Fresh fatty	898.92 ± 70.98 ^a^	256.41 ± 7.35 ^b^	940.34 ± 94.12 ^a^	<0.01
	9	methional M	916.4	407.933	1.08419	Musty potato	125.99 ± 337.31	133.32 ± 44.31	110.18 ± 25.38	0.661
	10	methional D	921.6	416.002	1.39781	Musty potato	167.08 ± 37.37	243.00 ± 12.92	—	0.029
	11	(E)-2-heptenal	918.2	410.679	1.25196	Fruity green	207.87 ± 6.21	289.93 ± 7.94	—	<0.01
	12	benzaldehyde M	980.5	519.474	1.15413	fruity cherry	2400.08 ± 30.65 ^b^	2348.76 ± 92.87 ^b^	2621.80 ± 64.76 ^a^	0.006
	13	benzaldehyde D	979.3	516.947	1.28859	fruity cherry	488.48 ± 21.42 ^c^	605.59 ± 32.95 ^b^	899.90 ± 7.51 ^a^	<0.01
	14	benzaldehyde T	980.1	518.652	1.46891	strong sharp	1157.42 ± 53.55	1234.71 ± 123.72	1312.12 ± 133.86	0.298
	15	Octanal M	1013.6	581.571	1.40125	orange peel	961.46 ± 52.94 ^a^	490.44 ± 8.27 ^b^	978.22 ± 81.28 ^a^	<0.01
	16	Octanal D	1012.8	580.234	1.82611	orange peel	317.81 ± 31.92 ^a^	121.10 ± 9.48 ^b^	334.52 ± 44.68 ^a^	<0.01
	17	2-octenal ( E)	1069.4	691.019	1.33185	fresh cucumber	170.20 ± 22.73 ^a^	69.02 ± 3.05 ^b^	48.03 ± 5.30 ^b^	<0.01
	18	n-Nonanal M	1104.7	770.689	1.47201	Orange fatty	1244.31 ± 73.52 ^a^	579.87 ± 13.80 ^c^	1090.93 ± 80.22 ^b^	<0.01
	19	n-Nonanal D	1104.4	770.007	1.50718	Orange fatty	412.58 ± 36.15 ^b^	242.85 ± 13.36 ^c^	508.71 ± 42.09 ^a^	<0.01
	20	n-Nonanal T	1103.9	768.833	1.95134	Orange fatty	180.96 ± 30.13 ^a^	55.28 ± 9.78 ^b^	143.80 ± 21.58 ^a^	0.001
Esters	21	methyl acetate	500.6	105.943	1.18735	Green etherial	113.52 ± 2.26 ^a^	117.49 ± 3.81 ^a^	68.22 ± 5.38 ^b^	<0.01
	22	ethyl 2-hydroxypropanoate	837.1	305.129	1.13711	sharp tart	—	—	34.18 ± 1.24	
	23	2-Hexen-1-ol M	853.2	323.333	1.18651	fruity green	395.99 ± 11.07 ^a^	207.55 ± 7.33 ^b^	199.65 ± 5.11 ^b^	<0.01
	24	2-Hexen-1-ol D	851.9	321.877	1.52421	fruity green	133.13 ± 8.03	120.65 ± 9.26	—	0.366
	25	ethyl 3-methylbutyrate	854.6	325.033	1.25525	fruity sweet	126.85 ± 6.00	201.40 ± 17.07	—	0.015
	26	Ethyl acetoacetate	919.9	413.216	1.599	fresh fruity	96.89 ± 1.69	186.44 ± 12.36	—	0.002
	27	methyl hexanoate	962.4	485.091	1.28844	fruity fatty	67.47 ± 3.15 ^b^	100.44 ± 6.73 ^b^	223.32 ± 21.88 ^a^	<0.01
	28	Methyl 2-furoate	962.8	485.778	1.15747	fruity mushroom	439.62 ± 39.06 ^ab^	499.30 ± 18.35 ^ab^	643.19 ± 69.03 ^a^	0.054
	29	isobutyl isovalerate	991.7	541.691	1.38165	sweet fruity	170.65 ± 8.05 ^b^	96.71 ± 6.33 ^c^	377.16 ± 19.65 ^a^	<0.01
	30	butyl 2-methylbutanoate	1046.1	643.088	1.3726	fruity tropical	53.25 ± 5.48	29.34 ± 1.57	—	0.014
	31	diethyl malonate	1070.8	694.023	1.24932	sweet fruity	437.74 ± 7.71 ^b^	578.92 ± 14.21 ^a^	174.23 ± 9.45 ^c^	<0.01
Alcohols	32	Isopropanol M	460.5	94.992	1.09257	alcohol musty	349.23 ± 11.82	380.01 ± 14.60	372.96 ± 8.58	0.240
	33	Ethanol	463.4	95.743	1.0544	strong alcoholic	1161.29 ± 43.50 ^ab^	1252.38 ± 18.68 ^a^	1069.51 ± 13.12 ^b^	0.011
	34	1-butanol M	665.2	165.902	1.18135	fusel oil	1595.46 ± 42.12 ^a^	1336.97 ± 4.94 ^b^	1566.99 ± 81.70 ^a^	0.026
	35	1-butanol D	666.4	166.444	1.37795	fusel oil	3341.67 ± 94.91	4721.94 ± 99.88	—	0.001
	36	2-Ethoxyethanol	739	213.137	1.0971	-	338.72 ± 12.99 ^b^	434.88 ± 13.90 ^a^	183.93 ± 9.12 ^c^	<0.01
	37	1-pentanol M	774.5	243.12	1.25023	fusel oil	797.03 ± 5.15 ^a^	638.31 ± 10.08 ^b^	641.65 ± 14.79 ^b^	<0.01
	38	1-pentanol D	772.9	241.702	1.4093	fusel oil	299.21 ± 28.04 ^a^	137.41 ± 16.11 ^b^	242.90 ± 27.82 ^a^	0.010
	39	1-pentanol T	773.2	241.966	1.50852	fusel oil	301.82 ± 8.05 ^a^	212.15 ± 0.71 ^b^	154.28 ± 7.45 ^c^	<0.01
	40	(Z)-3-Hexen-1-ol	874.3	349.051	1.22316	fresh green	125.73 ± 3.94	202.08 ± 3.90	—	<0.01
	41	n-Hexanol	880.9	357.439	1.32517	green fruity	91.01 ± 4.76 ^b^	109.44 ± 4.37 ^b^	160.51 ± 10.52 ^a^	0.001
	42	2-Heptanol	922.6	417.547	1.37468	fresh lemon	71.35 ± 11.40	110.68 ± 8.87	—	0.053
	43	1-Heptanol	962.7	485.59	1.38815	solvent-like	146.66 ± 7.55 ^a^	125.59 ± 1.32 ^b^	74.25 ± 6.65 ^c^	<0.01
Ketone	44	2-Propanone	494	104.05	1.12552	solvent ethereal	10025.42 ± 55.97 ^a^	9727.94 ± 46.71 ^b^	9711.63 ± 83.47 ^b^	0.023
	45	2-Butanone M	581	131.877	1.07811	acetone-like	2522.22 ± 22.85 ^a^	2031.50 ± 64.10 ^c^	2224.84 ± 16.02 ^b^	<0.01
	46	2-Butanone D	580.1	131.591	1.16366	acetone-like	603.57 ± 12.89 ^c^	798.66 ± 19.62 ^b^	850.49 ± 10.02 ^a^	<0.01
	47	2-Butanone T	581	131.888	1.24956	acetone-like	3713.07 ± 25.54 ^b^	4364.15 ± 78.26 ^a^	3293.61 ± 131.89 ^c^	<0.01
	48	2-pentanone	681.7	173.549	1.12621	Sweet fruity	445.79 ± 6.97	389.71 ± 11.76	—	0.015
	49	2-heptanone M	890.3	369.786	1.26005	Cheese fruity	593.24 ± 7.56 ^a^	611.82 ± 9.57 ^a^	272.62 ± 11.05 ^b^	<0.01
	50	2-heptanone D	889.1	368.161	1.63693	Cheese fruity	310.52 ± 3.14 ^b^	385.84 ± 8.38 ^a^	54.52 ± 4.15 ^c^	<0.01
	51	Cyclohexanone	899.7	383.019	1.15248	minty acetone	100.71 ± 0.66 ^b^	120.14 ± 2.75 ^a^	128.17 ± 3.32 ^a^	0.001
	52	1-octen-3-one M	962.6	485.408	1.26112	herbal mushroom	322.33 ± 27.37	42.38 ± 3.99	—	<0.01
	53	1-octen-3-one D	961.2	482.982	1.67736	herbal mushroom	852.58 ± 36.02 ^a^	291.64 ± 7.53 ^b^	146.48 ± 4.97 ^c^	0.001
Pyrazine	54	Methylpyrazine	836	303.923	1.08311	nutty cocoa	712.77 ± 5.93	961.00 ± 14.00	—	<0.01
	55	2,5-dimethylpyrazine M	925	421.354	1.11583	cocoa roasted nuts	289.22 ± 97.16	508.72 ± 214.08	—	0.002
	56	2,5-dimethylpyrazine D	922.3	416.987	1.50065	cocoa roasted nuts	3176.17 ± 8.49	4835.77 ± 34.11	—	0.003
	57	2-ethyl-3-methylpyrazine	1025.5	603.381	1.17094	nutty peanut	1611.58 ± 76.27	2470.46 ± 175.81	—	0.011
	58	2-ethyl-3,5-dimethylpyrazine	1090	736.361	1.23234	burnt almonds	319.11 ± 6.04	555.82 ± 42.56	—	0.005
Organic acid	59	hexanoic acid	1040.8	632.652	1.29394	sour fatty	41.30 ± 2.53 ^b^	38.66 ± 1.87 ^b^	51.56 ± 2.05 ^a^	0.013
Furan	60	2-ethylfuran	662.2	164.548	1.05125	solvent ethereal	—	—	438.41 ± 50.14	
	61	5-methylfurfuryl alcohol	962.5	485.317	1.57392	sweet caramellic	164.83 ± 2.90 ^a^	85.94 ± 2.29 ^b^	51.47 ± 5.73 ^c^	<0.01
Hydroxy ketone	62	3-hydroxy-2-butanone M	710.1	191.471	1.06034	sweet buttery	1751.54 ± 50.45 ^a^	932.73 ± 52.11 ^b^	1887.01 ± 36.97 ^a^	<0.01
	63	3-hydroxy-2-butanone D	708	190.028	1.3319	sweet buttery	1465.10 ± 77.40 ^a^	663.79 ± 13.39 ^b^	1445.86 ± 123.58 ^a^	0.001
	64	4-Hydroxy-4-methyl-2-pentanone	860.3	331.775	1.13711	-	87.08 ± 14.26 ^b^	98.89 ± 10.98 ^b^	167.15 ± 15.80 ^a^	0.013
Sulfur compounds	65	allyl isothiocyanate	905.5	391.422	1.09622	strong pungent	171.97 ± 12.09 ^b^	277.89 ± 5.37 ^a^	85.65 ± 2.88 ^c^	<0.01
	66	Furfuryl methyl sulfide	979.7	517.755	1.38815	onion garlic	121.97 ± 5.77 ^b^	165.86 ± 10.44 ^a^	101.04 ± 2.11 ^b^	0.002

^1^ The data in the same column marked with different letters showed significant differences (*p* < 0.05), while the data marked with the same letter or no letter showed no significant differences (*p* > 0.05).

**Table 7 molecules-28-00165-t007:** Sensory evaluation and scoring criteria of BTS meat under different processing methods.

Index		Scores
Appearance	Uniform size, no damage on the surface	13–20
	The size is relatively uniform, and the surface is slightly damaged	9–12
	Uneven size, serious surface damage	0–8
Colour	The surface is golden yellow, with uniform color and luster	13–20
	The surface is light yellow, with uneven color and luster	9–12
	The surface is burnt black, with uneven color and no luster	0–8
Texture	Crispy outside, tender inside, moderate hardness, juicy	13–20
	Crisp outside and tender inside, slightly hard or soft, more juice	9–12
	The meat is very dry, hard or soft, with little juice	0–8
Smell	Pure fragrance, no smell of mutton	13–20
	Average fragrance, a little mutton smell	9–12
	No fragrance, heavy mutton smell	0–8
Acceptability	Easy to accept	13–20
	Easier to accept	9–12
	Not easy to accept	0–8

## Data Availability

Not applicable.

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
