# Peer review of "Effect of Different Heat Treatments on the Quality and Flavor Compounds of Black Tibetan Sheep Meat by HS-GC-IMS Coupled with Multivariate Analysis"

_molecules, 2022, doi:10.3390/molecules28010165_

Round 1
Reviewer 1 Report
Dear Authors,
Detailed notes on the manuscript:
1) Methodology for statistical analysis; you use parametric ANOVA - add that the normality of the distribution of the data population was examined, the homogeneity of variance in the samples was examined (what tests, what result - and that it allowed the use of parametric ANOVA)
2) Figures are not made in accordance with the MDPI editorial standard
3) Figures no. 6, 8 are illegible (description)
4) Chapter 5. Conclusions is a summary, I suggest changing the title to "Summary"
Author Response
Dear Prof.:
Thank you very much for your letter of 13-December-2022 informing us the conditional acceptance of our manuscript, "Effect of different heat treatments on the quality and flavor compounds of Black Tibetan sheep meat by HS-GC-IMS coupled with multivariate analysis" (molecules-2101598), together with the highly valuable comments, suggestions, and instructions from the editor and reviewers. We went through the letter carefully and greatly appreciated the editor and reviewers for the constructive comments, helpful suggestions, and detailed instructions.
A point-to-point response to the comments, suggestions, and instructions from the editor and reviewers were involved in this letter as following.
Explanations, corrections/revisions, and investigations based on the comments and suggestions from Reviewer 1:
Comment 1: Methodology for statistical analysis; you use parametric ANOVA - add that the normality of thedistribution of the data population was examined, the homogeneity of variance in the sampleswas examined (what tests. what result - and that it allowed the use of parametric ANOVA)
Response: Thanks a lot for the comments and suggestions. Before the analysis of variance, we tested the normal distribution and homogeneity of variance of the data. P>0.05 for most of the data obeyed the test of normal distribution and homogeneity of variance. At the same time, we also checked the normal histogram of data, and the normal curve distribution of all data is symmetric bell curve. We are very sorry that the homogeneity of variance test of amino acid data is not very good, so in this part, we only listed the factual data, and did not conduct further joint analysis. The following table shows the normal distribution and homogeneity of variance test results of all our data.
Comment 2: Figures are not made in accordance with the MDPI editorial standard
Response: Thanks a lot for the comments and suggestions. We are sorry that the image does not conform to the MDPI editing standard. We have changed the lines of the table to 1 point and the DPI of the image to 300 according to the MDPI editing standard.
Comment 3: Figures no. 6, 8 are illegible (description)
Response: Thanks a lot for the comments and suggestions. We appologize for the blurring of the picture. According to your reminder, we reedited it with Photoshop to make the picture clearer.
Comment 4: Chapter 5.Conclusions is a summary, I suggest changing the title to "Summary"
Response: Thanks a lot for the comments and suggestions. According to your opinion, we have changed the title of Chapter 5 to "Summary" with red font.
We highly value this opportunity and we have endeavored to revise the manuscript according to the instructions, comments, and suggestions from the editor and reviewers, necessary explanations, and corrections/revisions have been done carefully to improve the quality of the manuscript. The changes in the revised manuscript and supporting information have been highlighted in different color.
We are very grateful to you for your kind advice, helpful instructions and continuous effort in the processing of our manuscript and we will be very happy to provide any further information if needed.
Yours sincerely,Xue Zhang and Lijuan Han, DrQinghai UniversityXining, China, 810000Tel: +86-18161335492; +86-15597460033E-mail: 1548488958@qq.com; hlj880105@163.com

Reviewer 2 Report
Dear authors,
Please consider the comments below to improve the manuscript.
L254 - Insert which type of oil was used for cooking to make sure that aromatic compounds of oil do not influence on meat aromatic compounds
Topic 4.21 - Did the authors use a specific statistical test to evaluate differences in samples among the panelists or the authors only considered the attributes?
Topic 2.1 - Explain for which reason the authors conducted this type of test.
L91 - Best sensory attributes in terms of what? For instance, on acceptability only boiling cooking was statistically different from the others . Which minimum score for a sample is considered to be ''acceptable''?
L143-146. Interesting! Insert the reason why the cooking methods affected the amino acid content. Does the temperature in each method provoked protein denaturation and promoting the release of amino acids? Include reports from literature and compare your results with other products.
L178-183 - Any idea why? Does the oil used for cooking influenced on this composition in fatty acids? Compare your results with literature.
Section 2.5.1, 2.5.2 and 2.5.3- Why the authors did not include the boiled samples? Are these data consistent with literature? Include information
Author Response
Dear Prof.:
Thank you very much for your letter of 13-December-2022 informing us the conditional acceptance of our manuscript, "Effect of different heat treatments on the quality and flavor compounds of Black Tibetan sheep meat by HS-GC-IMS coupled with multivariate analysis" (molecules-2101598), together with the highly valuable comments, suggestions, and instructions from the editor and reviewers. We went through the letter carefully and greatly appreciated the editor and reviewers for the constructive comments, helpful suggestions, and detailed instructions.
A point-to-point response to the comments, suggestions, and instructions from the editor and reviewers were involved in this letter as following.
Explanations, corrections/revisions, and investigations based on the comments and suggestions from Reviewer 2:
Comment 1: L254 - Insert which type of oil was used for cooking to make sure that aromatic compounds of oildo not influence on meat aromatic compoundsResponse: Thanks a lot for the comments and suggestions. According to your reminder, we inserted the cooking oil type in the manuscript L254-257 through the red font. L254-257:The edible oil used for pan-frying and deep-frying is soybean vegetable oil. The study found that soybean oil has fewer kinds of volatile substances and lower content, in which the content of hexanal, pentanal, (E) - 2-heptanal, (E) - 2-decenal and nonanal is higher. Thus, according to the previous study, we considered that aromatic compounds of oil do not influence on meat aromatic compounds. Comment 2: Topic 4.21 - Did the authors use a specific statistical test to evaluate diferences in samplesamong the panelists or the authors only considered the attributes?Response: Thanks a lot for the comments and suggestions. We did not use a specific statistical test to evaluate the sample differences among team members, but we took their attributes into consideration. All meat samples are cooked by the same person according to strict cooking time and temperature. At the same time, for the evaluators, the four groups of meat samples have obvious differences in appearance, color, texture and flavor, which are unique to this cooking method. Comment 3: Topic 2.1 - Explain for which reason the authors conducted this type of test.Response: Thanks a lot for the comments and suggestions. We use sensory evaluation because the quantitative indicators usually measured by chemical methods cannot explain the overall situation of a sensory evaluation well, and chemical detection cannot fully explain the interaction of various sensory elements. An evaluation of meat flavor is essential to determining whether or not this product will be accepted, sensory evaluation is an important tool in a product development. In this study, sensory evaluation can directly reflect people's preference for BTS meat with different heat treatments. Comment 4: L91 - Best sensory attributes in terms of what? For instance, on acceptability only boilingcooking was statistically different from the others. hich minimum score for a sample isconsidered to be "acceptable"?Response: Thanks a lot for the comments and suggestions. Our sensory evaluation refers to the evaluation of the appearance, color, texture, aroma and acceptability of meat samples by sensory evaluation members according to the scoring standards listed in Table 7. Each index is divided into three levels: best, better and bad. Our sensory evaluation and scoring standards refer to the methods of Neill. The higher the sensory evaluation score, the better the sample conforms to the standard; The lower the score, the closer to the poor standard. For example, in terms of acceptability, the boiling sample fraction is at least 14.1, greater than 8. It indicates that the evaluator can accept boiled samples. But the other three samples are preferred.1.O’Neill, C.M.; Cruz-Romero, M.C.; Duffy, G.; Kerry, J.P. Comparative Effect of Different Cooking Methods on the Physicochemical and Sensory Characteristics of High Pressure Processed Marinated Pork Chops. Innovative Food Science & Emerging Technologies 2019, 54, 19–27, doi:10.1016/j.ifset.2019.03.005. Comment 5: L143-146.Interesting! lnsert the reason why the cooking methods affected the amino acidcontent. Does the temperature in each method provoked protein denaturation and promoting therelease of amino acids? lnclude reports from literature and compare your results with otherproductsResponse: Thanks a lot for the comments and suggestions. According to your opinion, we added the reason why cooking methods affect the amount of amino acids in the amino acid discussion. L398-L409: Lopes et al. [33] found that the retention rate of TAA in grilling beef was higher than that in microwave, and boiling cooked beef. Wilkinson et al. [34] showed that the longest muscle sample of pork back boiled for 90 min at 60-75 ℃ had an amino acids retention rate of lower than 90%; however, the retention rate increased at a lower cooking temperature. Nyam et al. [35] found that the longer the chicken breast was heat treated at the same temperature, the more amino acids was lost. In this study, the amino acids content of cooked BTS meat was significantly reduced, and the amino acids content of pan-fried BTS meat was the highest. This may be because the protein denaturation of BTS meat at a higher temperature (226-228 ℃) increases the release of amino acids, and the increase of water loss leads to the increase of amino acids retention rate [33]. But the boiling time is long (60 min), which leads to the loss of amino acids in water.
[33] Lopes, A.F.; Alfaia, C.M.M.; Partidário, A.M.C.P.C.; Lemos, J.P.C.; Prates, J.A.M. Influence of Household Cooking Methods on Amino Acids and Minerals of Barrosã-PDO Veal. Meat Science 2015, 99, 38–43, doi:10.1016/j.meatsci.2014.08.012.
[34]Wilkinson, B.H.P.; Lee, E.; Purchas, R.W.; Morel, P.C.H. The Retention and Recovery of Amino Acids from Pork Longissimus Muscle Following Cooking to Either 60℃ or 75℃. Meat Science 2014, 96, 361–365, doi:10.1016/j.meatsci.2013.07.019.
[35]Nyam, K. L.; Goh, K. M.; Chan, S. Q.; Tan, C. P.; Cheong, L. Z., Effect of sous vide cooking parameters on physicochemical properties and free amino acids profile of chicken breast meat. Journal of Food Composition and Analysis 2023, 115, 105010. doi:10.1016/j.jfca.2022.105010.Comment 6: L178-183 - Any idea why? Does the oil used for cooking influenced on this composition in fattyacids? Compare your results with literature.Response: Thanks a lot for the comments and suggestions. According to your reminder, we have added an explanation why the MUFA and PUFA content in pan-fried and deep-fried BTS meat is significantly higher than that in boiled and baked BTS meat in L435-L442 discussed about fatty acids.L435-442: Haak et al. [43] found that the content of SFAs and UFAs in oils rich in unsaturated fatty acids would be significantly reduced after cooking. The previous research of the research group found that [44], BTS meat is a high-quality raw material with UFAs content significantly higher than SFAs content. Ge et al. [45] found that MUFAs and PUFAs in pan-fried and fried beef and pork are higher than those in boiled and roasted beef and pork. In this study, the content of MUFAs and PUFAs in pan-fried and deep-fried BTS meat was significantly higher than that in baked and boiled BTS meat, this was consistent with the research results of Ge et al. [45].[43] Haak, L.; Sioen, I.; Raes, K.; Camp, J. V.; Smet, S. D., Effect of pan-frying in different culinary fats on the fatty acid profile of pork. Food Chemistry 2007, 102, (3), 857-864. doi:10.1016/j.foodchem.2006.06.054.[44] Zhang, X.; Han, L.; Hou, S.; Raza, S. H. A.; Wang, Z.; Yang, B.; Sun, S.; Ding, B.; Gui, L.; Simal-Gandara, J.; Shukry, M.; Sayed, S. M.; Al Hazani, T. M. I., Effects of different feeding regimes on muscle metabolism and its association with meat quality of Tibetan sheep. Food Chemistry 2022, 374, 131611. doi: 10.1016/j.foodchem.2021.131611. [45] Ge, X.; Zhang, L.; Zhong, H.; Gao, T.; Jiao, Y.; Liu, Y., The effects of various Chinese processing methods on the nutritional and safety properties of four kinds of meats. Innovative Food Science & Emerging Technologies 2021, 70, 102674. doi:10.1016/j.ifset.2021.102674.
Comment 7: Section 2.5.1, 2.5.2 and 2.5.3- Why the authors did not include the boiled samples? Are these data consistent with iterature? nclude information
Response: Thanks a lot for the comments and suggestions. The reason why we didn't add boiled samples to the flavor is we sent samples for flavor testing of cooked BTS meat, but the testing company told us that the content of flavor substances in cooked BTS meat was very low, and there was no comparability with the other three samples. The flavor substances in BTS meat cannot be compared with the other three samples for the analysis of the same kind of substances, and the fingerprint cannot be analyzed. At the same time, we consider that the cooked BTS meat is lower than the other three samples in sensory, edible and nutritional aspects in the previous study. Therefore, we did not consider cooking in terms of flavor.
We highly value this opportunity and we have endeavored to revise the manuscript according to the instructions, comments, and suggestions from the editor and reviewers, necessary explanations, and corrections/revisions have been done carefully to improve the quality of the manuscript. The changes in the revised manuscript and supporting information have been highlighted in different color.
We are very grateful to you for your kind advice, helpful instructions and continuous effort in the processing of our manuscript and we will be very happy to provide any further information if needed.
Yours sincerely,Xue Zhang and Lijuan Han, DrQinghai UniversityXining, China, 810000Tel: +86-18161335492; +86-15597460033E-mail: 1548488958@qq.com; hlj880105@163.com
